# Young Novice Drivers’ Cognitive Distraction Detection: Comparing Support Vector Machines and Random Forest Model of Vehicle Control Behavior

**DOI:** 10.3390/s23031345

**Published:** 2023-01-25

**Authors:** Qingwan Xue, Xingyue Wang, Yinghong Li, Weiwei Guo

**Affiliations:** 1Beijing Key Laboratory of Urban Intelligent Traffic Control Technology, North China University of Technology, Beijing 100144, China; 2Engineering Research Center of Catastrophic Prophylaxis and Treatment of Road & Traffic Safety of Ministry of Education, Changsha University of Science & Technology, Changsha 410114, China

**Keywords:** young novice driver, distraction driving, vehicle control behavior, support vector machines, random forest model

## Abstract

The use of mobile phones has become one of the major threats to road safety, especially in young novice drivers. To avoid crashes induced by distraction, adaptive distraction mitigation systems have been developed that can determine how to detect a driver’s distraction state. A driving simulator experiment was conducted in this paper to better explore the relationship between drivers’ cognitive distractions and traffic safety, and to better analyze the mechanism of distracting effects on young drivers during the driving process. A total of 36 participants were recruited and asked to complete an n-back memory task while following the lead vehicle. Drivers’ vehicle control behavior was collected, and an ANOVA was conducted on both lateral driving performance and longitudinal driving performance. Indicators from three aspects, i.e., lateral indicators only, longitudinal indicators only, and combined lateral and longitudinal indicators, were inputted into both SVM and random forest models, respectively. Results demonstrated that the SVM model with parameter optimization outperformed the random forest model in all aspects, among which the genetic algorithm had the best parameter optimization effect. For both lateral and longitudinal indicators, the identification effect of lateral indicators was better than that of longitudinal indicators, probably because drivers are more inclined to control the vehicle in lateral operation when they were cognitively distracted. Overall, the comprehensive model built in this paper can effectively identify the distracted state of drivers and provide theoretical support for control strategies of driving distraction.

## 1. Introduction

According to the statistics of the World Health Organization, traffic accidents have become the ninth leading cause of death, and approximately 1.25 million people die from traffic accidents globally every year [1]. Among the numerous factors leading to road accidents, driver attention and distraction has been identified as one of the major contributors [2,3]. According to the statistics of the National Safety Administration [4], 3477 people were killed and more than 391,000 were injured in vehicle accidents caused by distracted drivers in the United States in 2015. The Texas Department of Transportation (TxDOT) reported that 46.8% of crashes in San Antonio, Texas, were classified as “distracted driving crashes” in 2018 [5]. Due to the increasing use of mobile phones and other intelligent in-vehicle devices, driving distraction has become more and more popular, especially in young drivers, and it was found that drivers in the 19–26 age group in China have the highest traffic accident rate [6].

Driving distraction can be defined as a “diversion of attention away from activities critical for safe driving toward a competing activity” [7,8]. Previous studies have categorized distraction into four types, i.e., visual distraction, cognitive distraction, auditory distraction, and action distraction [9,10], of which the main forms are visual distraction and cognitive distraction. Visual distraction means that the driver’s vision deviates from the road ahead for a period of time or frequently deviates from the road ahead during a certain period of time, which is called “eyes off road” [11]. Cognitive distraction means that the human brain is thinking about things that have nothing to do with driving, but the sight does not leave the road ahead, which is called “mind off road” [12]. Compared visual distraction, cognitive distraction is more difficult to recognize because the signs of cognitive distraction are usually not readily apparent, which is called “see but not see”.

To avoid crashes induced by distraction, adaptive distraction mitigation systems which can provide assistance to reduce distraction state have been developed [13]. For this system, the key is how to detect a driver’s distraction state. Indicators including driving performance [14], eye-movement characteristics [15], and even psychological signals [16] haven been adopted to detect a driver’s cognitive distraction. The effect of cognitive distraction on driving performance has been mainly conducted from two aspects: the lateral control behavior and longitudinal control behavior of drivers. Generally, drivers prefer to increase their steering wheel reversal rates and steering wheel angular velocities [17,18], and reduce lane offset distances [19] for cognitive distraction conditions. Most studies believe that during cognitive distraction, the driver’s vision is mainly focused on the center of the road ahead [20,21], resulting in the driver’s enhanced perception of lane offset and thus correcting the amount of lane offset by frequently operating the steering wheel [22], which is assumed as kind of compensation behavior. However, it seems that researchers have not met an agreement regarding the effect of cognitive distraction on drivers’ longitudinal control behavior. Horrey et al. [23] found that drivers were more inclined to increase the headway distance from the lead vehicle when performing a cognitively secondary driving task. However, the increased distance induced by cognitive distraction has not been found in other studies [19]. Another divergence is speed changes. Drivers were found to drive faster when they were cognitively distracted as they concentrated on road center [24], while a reduced driving speed was found for cognitive distraction in other studies [25,26].

Bio-psychological signals and eye movement data have been adopted to detect drivers’ cognitive distraction. Commonly used bio-psychological indicators include an electroencephalogram [27], electrocardiography [28], and skin conductivity [29]. The eye movement feature mainly refers to the obvious changes in the driver’s eye saccade, gaze, blinking, and other behaviors under the distracted driving state [30]. However, to compare different indicators of driving distraction based on cardiac physiological indicators and eye movement indicators, invasive detection methods must be used including the use of eye movement instruments, electrocardiographs, electrode caps and other equipment. These testing instruments are bulky and the driver needs to wear them on the body where they can become invasive for the driver. Wearing them for a long time is likely to cause physical discomfort, which will cause some interference for the driver. The detection process is easily affected by the external environment resulting in abnormal data, and wearing the equipment for a long time is likely to cause physical discomfort which will cause some interference for the driver. The detection process is easily affected by the external environment and this may lead to abnormal data.

Driving performance data are the vehicle motion state data output by the driving simulator itself, which can characterize the driver’s control ability to manipulate the vehicle during the distraction process. As a non-invasive detection index, it has gradually become a research hotspot in detecting driving distraction in recent years. Establishing a real-time accurate driver distraction discrimination model is the premise and core of a driver distraction warning system. In the field of driver distraction discrimination, the more commonly used and mature discrimination models include various combinations of algorithms, such as support vector machines [28], random forests [31], neural networks [32], and hybrid network models [33]. The SVM model was based on supervised learning, and it has been widely adopted in the field of pattern recognition [34], financial engineering [35,36] and automotive engineering [37], with the advantages in solving problems with small samples and high-dimensional and non-linear datasets [38]. The random forest model incorporates the idea of random subspaces and the bagging method, which is robust to noise and outliers in the data [39] by building multiple decision trees and then merging them together to obtain more accurate and stable prediction results.

Most of the available distracted driving literature has been conducted in the form of comprehensive distractions, with fewer studies on the discrimination of cognitive distractions alone and fewer studies specifically on distractions in young drivers. Therefore, this paper aims to (1) take the laboratory driving simulator of North China University of Technology as the research platform and young drivers as the research object to design a cognitive distracted driving experiment; (2) carry out one-way analysis of variance on the index data of normal driving and distracted driving, and extract feature data with significant differences; and (3) establish an SVM discriminant model and random forest discriminant model for the extracted index data, and compare the performance differences of different models.

## 2. Methods

### 2.1. Apparatus

The North China University of Technology driving simulator (NCUT Sim) was used for this experiment (as shown in Figure 1). The hardware system is composed of a cockpit and the annular display screen. The cockpit includes the dashboard, steering wheel, seat, automatic gearbox and other components, which are in full accordance with a real vehicle. Three 46-inch-wide LCD screens are adopted to present the driving environment for the participants. The road scene modeling is built by the 3D virtual reality simulation software UC-win/RoadVer.14.1.0, and simulated driving scenarios are projected on the display screen (with a resolution of 1920 × 1080 pixels) through the control center directly. During the experiment, more than 60 vehicles and related road parameters such as speed, acceleration, accelerator, and brake pedal opening can be collected simultaneously, with the sampling frequency at 50 Hz.

### 2.2. Scenario Design

This experiment simulates the road environment of a 2-way, 4-lane urban expressway, with a central barrier, lane width of 3.5 m. There is no pedestrian lane and a non-motorized lane on both sides of the road according to the “Urban Road Engineering Design Specification”. The speed limit of the road is 80 km/h, where the speed set by the lead vehicle can reach up to 72 km/h. As shown in Figure 2, the road section is divided into 2 parts, the normal driving section and the cognitive distraction section. The length of the normal driving section is 4 km. At the beginning of the experiment, the lead vehicle was stationary ahead of the driver, and when the participant was 50 m behind the lead vehicle, the lead vehicle began to accelerate at 2 m/s^2^ until it reached 72 km/h. In each drive, the lead vehicle would drive at a constant speed of 72 km/h and the main vehicle stopped at the center of the outermost lane of the road at a distance of 50 m. The lead vehicle began to accelerate at an acceleration of 2 m/s^2^, and when the speed reached 72 km/h, it began to drive at a constant speed of 72 km/h. The lead vehicle would brake 4 times with the deceleration rate of 4 m/s^2^ once it passed the predesigned deceleration points. For the normal driving section, participants were asked to drive as they normally would.

### 2.3. Secondary Tasks

To investigate young novice drivers’ driving performance induced by cognitive distraction, numerous methods have been adopted in previous studies, e.g., math problems [40], digital reading of roadside signs [41], and n-back memory tasks [42,43]. The n-back task requires participants to respond verbally to a delayed digit recall task, and it allows for setting different difficulty levels of cognitive load [12]. This method was adopted. In this experiment, 2 levels of difficulty were designed, i.e., 1-back and 2-back. The 1-back task requires participants to memorize the digits and recall the number 1 back in the sequence; the 2-back task requires participants to memorize the current digits and recall the number two back in the sequence. For one cognitive task, a total of 10 digits would play randomly with an interval of 1.5 s. Figure 3 shows an example of the n-back task adopted in this experiment.

### 2.4. Participants and Procedure

A total of 36 participants were recruited for this experiment. The average age of participants was 24, ranging from 22 to 26 years old (mean = 24, S.D. = 0.92). Each participant held a valid driver license and had at least 1 year of driving experience. After arrival, each participant was briefed on the requirements of the experiment. They were asked to drive as they normally would and sign an informed consent form. A 15-min pre-test, including both normal driving and n-back test, would be conducted before the formal experiment. For the formal experiment, each participant had to drive 3 times, each test scenario was conducted separately, and the order of driving scenarios were counterbalanced. They could request to stop the experiment anytime if he/she felt any discomfort during the formal experiment. All participants received 50 RMB (around 7.75 USD) for their participation in the study.

### 2.5. Model Construction

#### 2.5.1. Support Vector Machines

The SVM model has significant advantages in solving nonlinearly separable classification problems. It can maximize the splitting edge between 2 types of data by constructing a multidimensional decision surface to accurately separate 2 types of sample data. Figure 4 illustrates the flow chart of cognitive distraction detection based on the SVM model. For this study, human cognition can seldom be represented by a linear model, and the SVM method maps the sample data into a high-dimensional space by introducing penalty parameters and penalty functions. The kernel function is adopted to achieve linearly divisible or approximately linearly divisible purposes. The radial basis function (RBF) was chosen as the kernel function for the SVM models with the advantages of reducing numerical difficulties and obtaining more robust results than other kernels [44]. Equation (1) shows the kernel function.
(1)$$K(x_{i},x_{j})=e^{-r||x_{i}-x_{j}|{|}^{2}}$$
in which, *x_i_* and *x_j_* represent 2 data points, and γ is a predefined positive parameter. By using the RBF kernel function, a nonlinear mapping of the sample data can be achieved. Driving performance metrics obtained from a driving simulator experiment were divided into training dataset and testing dataset with a proportion of 8:2. In training, 3 parameter-searching algorithms, i.e., Genetic Algorithm (GA), Particle Swarm Optimization algorithm (PSO), and Cuckoo Search algorithm (CS) were adopted to find the best parameterization. Detailed parameter search ranges for each algorithm are shown in Table 1.

#### 2.5.2. Random Forest Model

Random forest is an integrated learning method based on statistical learning theory and can be used as a classifier [45]. It is an extension of the decision tree. However, only 1 tree may reduce the accuracy of the classification model. Thus, random forest compensates the insufficiency by establishing a forest [46]. The bootstrap sampling method is employed by random forest to extract different training datasets with data putback from the original dataset. Each training dataset is used to generate a decision tree. The method of counting votes is adopted to record the output category of each decision tree [47], as shown in Figure 5.

In this study, data were firstly labelled with −1 as the distracted driving and 1 as the normal driving. Data were divided into training dataset and testing dataset with the same proportion of SVM. For the random forest method, the number of decision trees has a direct effect on the computational efficiency and classification results. The efficiency of the algorithm may decrease if there are too many decision trees, while the classification accuracy may decrease with the decrease in decision trees [39]. Thus, a grid search was used in this paper to find the best parameterization. The random forest construction of this study is illustrated in Figure 6.

### 2.6. Indicators for Cognitive Distraction Detection

A variety of driving performance metrics have been proposed for drivers’ cognitive distraction detection, among which, driving speed, acceleration, lane deviation, and steering wheel angle were the most adopted ones [48]. In this paper, relevant indicators were extracted from both lateral lane-keeping performance and longitudinal acceleration performance (as shown in Table 2). ANOVA was first adopted to test the difference induced by a secondary task, and we set the significance level of the one-way ANOVA at 0.05. Only indicators which were found to be significantly affected by the secondary task were inputted into the 2 detection models. A suitable time window can help to improve the detection accuracy and the accuracy of the detection model will be reduced by a time window that is too short [49]. Thus, a time window of 2 s was set from 2 s ahead of the lead vehicle’s brake onset, with an overlap of 1.5 s.

## 3. Analysis of Significant Differences in Distraction Indicators

For the extracted dataset, this paper adopts the method of single-factor analysis of variance to carry out statistical analysis of the significance of the extracted driving performance data. The statistical results showed that there are six indicators which show significant differences between normal driving and cognitive distracted driving, and the results of the analysis are shown in Table 3.

### 3.1. Lateral Driving Performance

#### 3.1.1. Steering Wheel Angle

The steering wheel angle refers to the angle at which the steering wheel deviates from left to right during driving, and the magnitude of its rotation angle reflects the stability of the participants’ lateral vehicle control. As shown in Table 3, where a significant difference caused by task demand was found in the percentage of steering wheel turns above the 75% quantile (F = 73.486, *p* < 0.05). Figure 7 shows that the cognitive task produced a larger steering wheel angle than the baseline and that the steering wheel angle followed as the difficulty of the cognitive task increased.

#### 3.1.2. Steering Wheel Speed

Steering wheel speed is the speed at which the steering wheel turns per second, and the driver’s operation of the steering wheel is directly reflected in the lateral fluctuations of the vehicle, reflecting the effectiveness of the driver’s lateral control of the vehicle.

As shown in Table 3, a significant difference caused by task demand was found in the percentage of steering wheel speed above the 75% quantile (F = 16.816, *p* < 0.05). Figure 8 shows that the cognitive task produced greater steering wheel speed than the baseline and that steering wheel speed increased with increasing difficulty in the cognitive task.

### 3.2. Longitudinal Driving Performance

#### 3.2.1. Longitudinal Speed

The average value of longitudinal speed represents the magnitude of the longitudinal speed of the vehicle during the subjects’ driving, as shown in Table 3. The cognitive distraction produced a significant difference in the average value of longitudinal speed of the vehicle (F = 5.457, *p* = 0.004 < 0.05). Figure 9 shows that the average value of longitudinal speed was higher when driving normally than when driving cognitively distracted.

#### 3.2.2. Headway Distance

The mean value of the headway distance refers to the safe distance between the main vehicle and the preceding vehicle during the following process, the standard deviation reflects the participants’ ability to control the safety distance when following a car.

As shown in Table 3, significant differences caused by task demand were found for both mean (F = 39.618, *p* < 0.05) and standard deviation (F = 10.592, *p* < 0.05) of headway distance. Figure 10 shows that the average headway distance during cognitively distracted driving is lower than normal driving.

#### 3.2.3. Throttle Angle

The standard deviation of throttle angle is the standard deviation of throttle pedal position during driving, which reflects the participants’ handling of longitudinal acceleration and deceleration of the vehicle.

As shown in Table 3, significant differences caused by task demand were found for standard deviation (F = 13.836, *p* < 0.05) of throttle angle, but no significant difference between two levels of task difficulty was found. As can be seen from Figure 11, the standard deviation of throttle angle is positively correlated with the difficulty of the cognitive distraction task.

## 4. Drivers’ Cognitive Distraction Detection

### 4.1. SVM Model Classification Results

Figure 12 shows the parameter search results from the three optimization algorithms. Among the three optimization algorithms, GA provides the best accuracy is 93.96% when c = 32.19, g = 4.38 compared with the other two methods. The cross-validation accuracy of the PSO-SVM model is 93.16% when c = 2.11, g = 3.74 and the optimal classification accuracy of the CS-SVM model is 92.45% when c = 6.21, g = 2.44.

The parameters c and g obtained by optimizing were re-entered into the model for training and prediction, the final classification accuracy of the final model was obtained as shown in Table 4.

### 4.2. Random Forest Classification Results

As shown in Figure 13, the model is trained by exhausting multiple super-parameter combinations, finding the optimal super-parametric combination according to the result of cross-validation for K = 5. It can be seen from Figure 9 that when the number of decision trees is 550 and the maximum number of features is five, the discrimination accuracy of the model can reach 91.91%, and the model can effectively discriminate the distracted state during driving.

### 4.3. Comparison of SVM and RF Classification Results

To measure the performance and the detection accuracy of the two methods, four indicators which have been widely adopted in previous studies [50,51] were then adopted in this paper, i.e., accuracy, precision, recall, and F_1_ score. The four indicators were defined as:(2)$$Accuracy=\frac{TP+TN}{TP+TN+FN+FP}$$
(3)$$Precision=\frac{TP}{TP+FP}$$
(4)$$Recall=\frac{TP}{TP+FN}$$
(5)$$F_{1}Score=\frac{2\times Precision\times Recall}{Precision+Recall}$$
in which, *TP* refers to true positive, which is correctly identified as normal driving; *FN* means false negative, which is falsely detected as distracted driving; *FP* means false positive, which is falsely detected as normal driving; and *TN* means true negative, which is correctly identified as distracted driving. Table 5 provides the detailed comparison results. Generally, SVM provides a better classification method. It was found that GA-SVM achieved 93.78% in terms of the average accuracy. Precision, recall, and F1 score are higher (95%, 86.9%, 90.7%) than the other three methods.

## 5. Discussion and Conclusions

In this paper, we analyzed the relationship between driving distraction and road safety by designing a virtual distracted driving simulation scenario experiment, with the following main findings and conclusions:(1)To investigate the effect of cognitive distraction on the ability of young drivers to control the vehicle in both lateral and longitudinal directions during driving, drivers were provided with two different cognitive loads through an n-back task, and a one-way ANOVA was used to analyze the significance of the extracted feature data, yielding differences in driving performance data between the cognitive load and baseline conditions. For lateral driving performance, consistent with previous studies [18,52], the mean and standard deviation of steering wheel speed became gradually larger as the cognitive load increased and the level of distraction increased. This indicates that drivers were influenced by cognitive distraction during driving and adopted the compensatory behavior of frequently correcting the steering wheel to reduce lateral fluctuations of the vehicle in order to increase lateral safety of the vehicle, and this compensation behavior is positively correlated with the degree of cognitive distraction.(2)In terms of longitudinal driving performance, as the difficulty of cognitive dis-traction increased, the standard deviation of the gas pedal increased, the mean value of longitudinal speed tended to decrease, and the mean and standard deviation of headway time distance increased. This indicates that participants’ ability to maneuver the vehicle longitudinally was affected by cognitive distraction, and the safety of vehicle longitudinal following was ensured by adopting frequent control of the gas pedal and reducing the speed, which made the vehicle’s following stability becomes less stable and the headway time distance becomes larger, which remains consistent with previous studies [23,25,26].(3)The data of indicators with significant differences between normal driving states and distracted driving states were divided into two categories: horizontal only and vertical only. A combination of horizontal and vertical indicators were selected as the input of the model, and SVM and random forest discriminant models were used to identify distracted states, respectively, where the SVM model used three different algorithms to optimize the parameters of the model parameters c and g. The results showed that the parameter-optimized SVM model outperforms the random forest model in terms of accuracy, precision, recall, F1 value, and other model performance metrics, with the genetic algorithm having the best parameter optimization. For both lateral and longitudinal indicators, the recognition effect of lateral indicators is better than that of longitudinal indicators, which also indicates that the driver is more inclined to control the lateral operation control of the vehicle in the process of cognitive distraction.

Overall, the comprehensive distraction discrimination model established in this paper can effectively identify the cognitive distraction status of drivers, and the discrimination results of the model can be used as theoretical support for later distracted driving monitoring and early warning and accident prevention systems. The findings of this paper have the following implications in practical engineering applications:(1)It provides theoretical support for the study of the intrinsic correlation mechanism between distracted driving and driving behavior operations.(2)It facilitates targeted detection of driver attention and research on distracted warning systems.(3)It provides model support for intelligent assisted driving and vehicle safety technology.

As this distracted driving simulation experiment was conducted under good traffic conditions, clear weather, and low external interference, the model was able to achieve the expected research objectives. Although it can achieve the expected research purpose, there is still a certain deviation compared with actual traffic scenarios, and the applicability in the real environment needs to be further studied. Therefore, the next step should be to carry out research on the discrimination of driving distractions under natural driving conditions, and to investigate the influence mechanism of cognitive distraction on drivers’ eye movements and heart physiology in terms of distraction indexes, so as to provide a more comprehensive theoretical basis for the identification of drivers’ driving states.

## Figures and Tables

**Figure 1 sensors-23-01345-f001:**
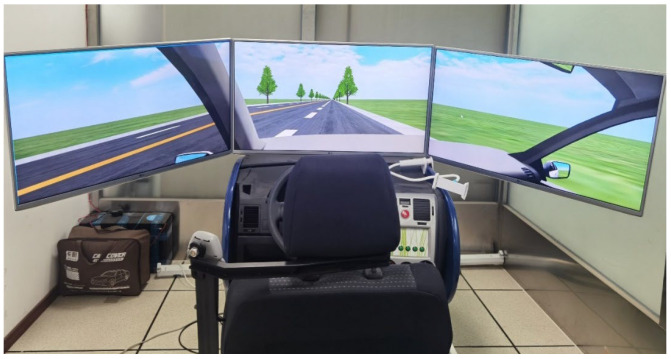
Driving simulation platform of North China University of Technology.

**Figure 2 sensors-23-01345-f002:**
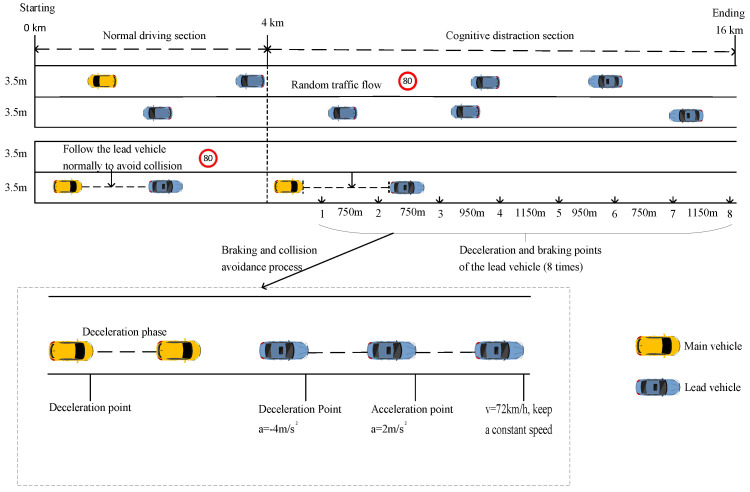
Schematic diagram of the scene.

**Figure 3 sensors-23-01345-f003:**
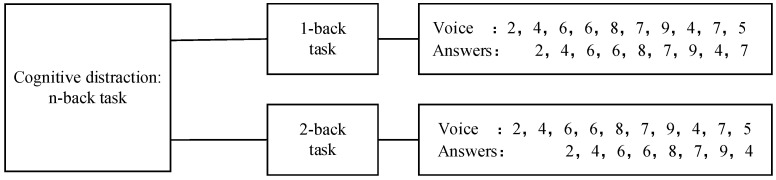
Schematic diagram of the n-back memory task.

**Figure 4 sensors-23-01345-f004:**
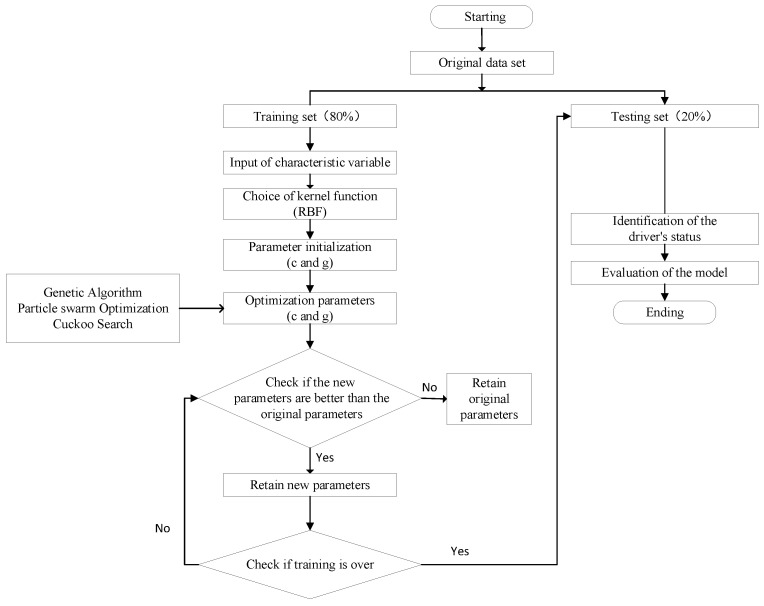
Flow chart of cognitive distraction detection based on the SVM model.

**Figure 5 sensors-23-01345-f005:**
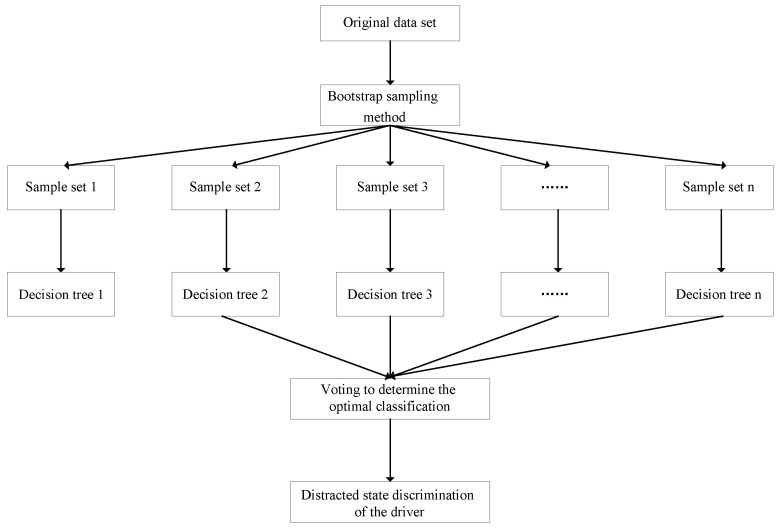
The flowchart of the random forest method.

**Figure 6 sensors-23-01345-f006:**
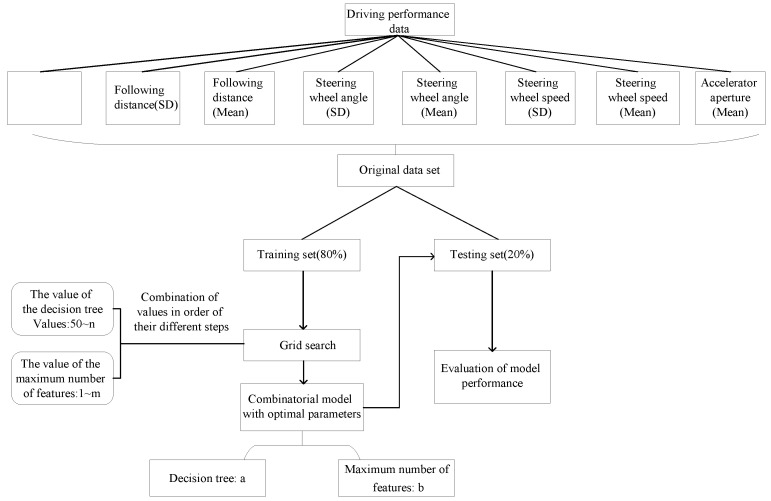
Construction of random forest model.

**Figure 7 sensors-23-01345-f007:**
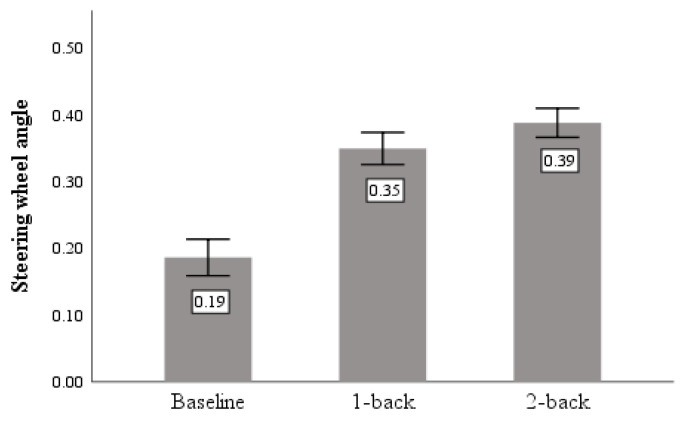
Steering wheel angle.

**Figure 8 sensors-23-01345-f008:**
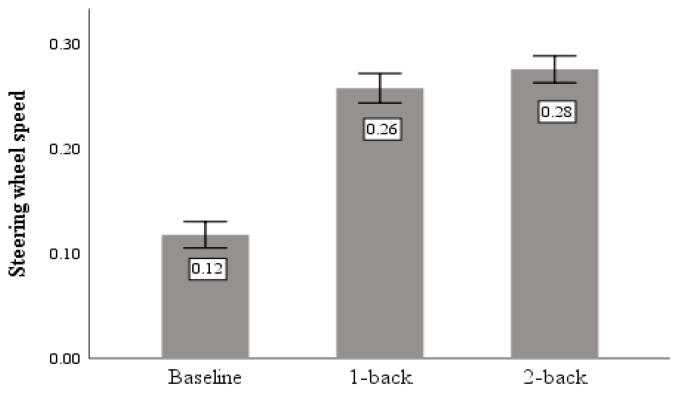
Steering wheel speed.

**Figure 9 sensors-23-01345-f009:**
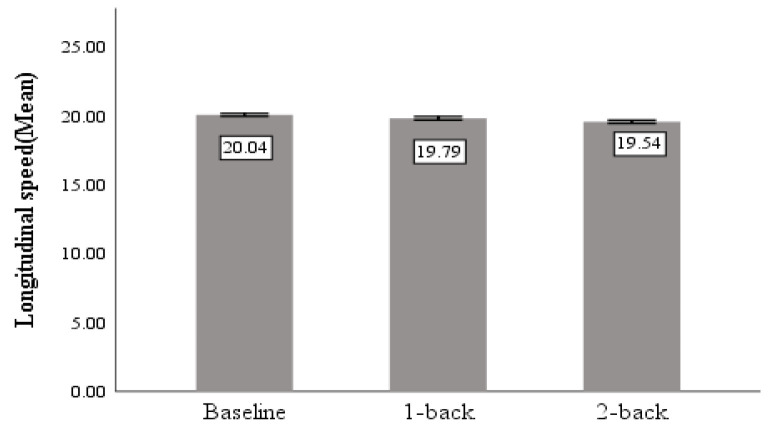
Mean value of longitudinal speed.

**Figure 10 sensors-23-01345-f010:**
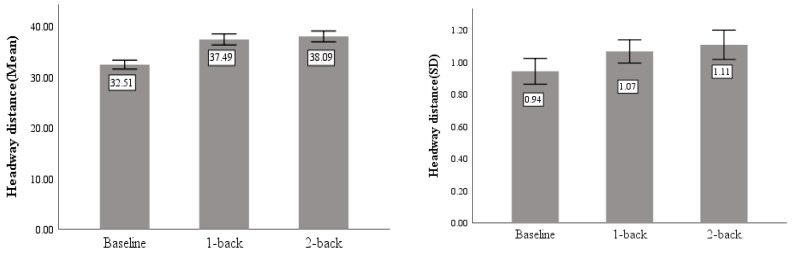
Mean and standard deviation of headway distance.

**Figure 11 sensors-23-01345-f011:**
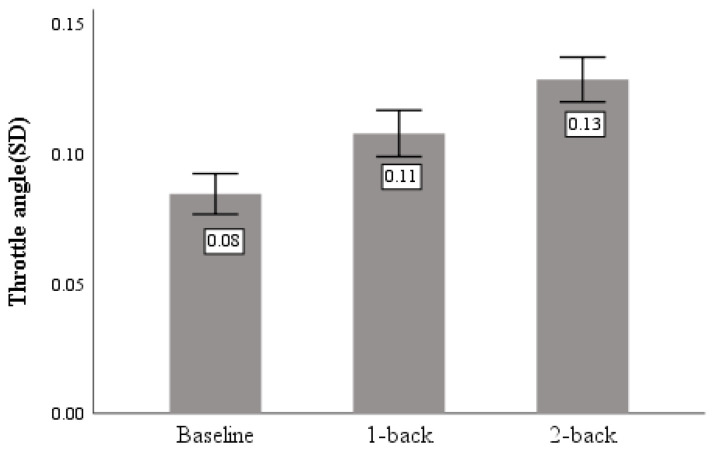
The standard deviation of throttle angle.

**Figure 12 sensors-23-01345-f012:**
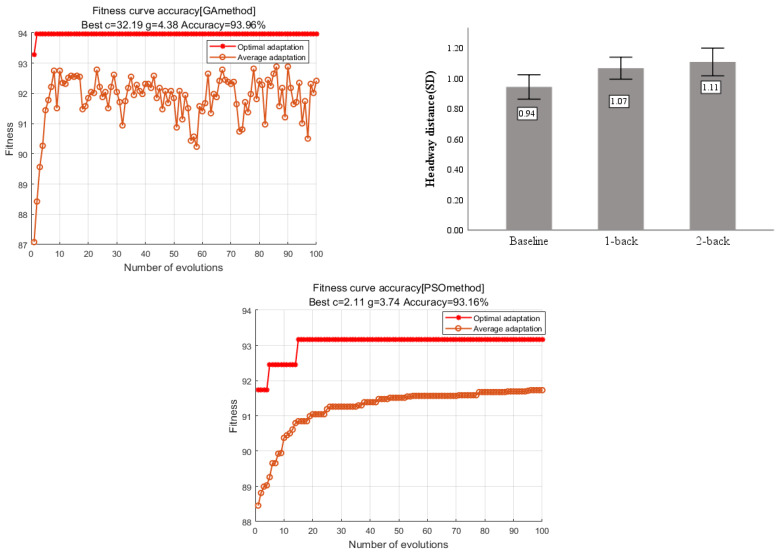
The fitness curve of the SVM model under three optimization algorithms.

**Figure 13 sensors-23-01345-f013:**
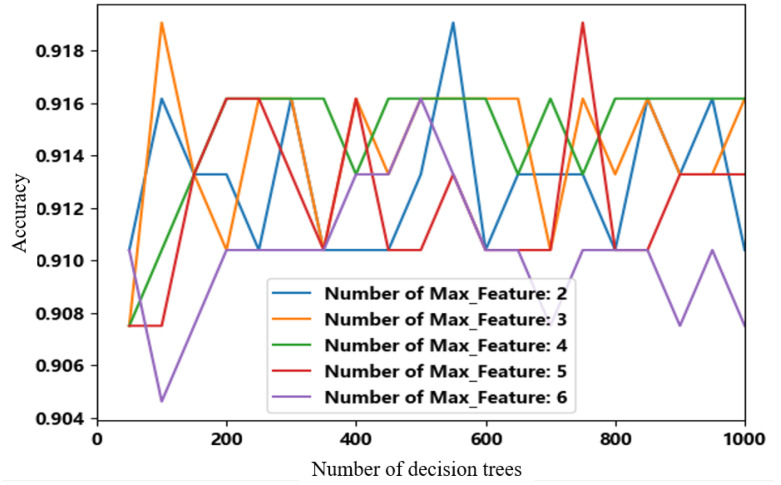
Line graph of optimal parameter combination search.

**Table 1 sensors-23-01345-t001:** Parameters search range.

	Range of Parameter c	Range of Parameter g	Number of Cross-Validation	Maximum Number of Iterations	Number of Populations	Probability of Crossover	Probability of Being Discovered by the Host	Probability of Variation
GA	[0, 100]	[0, 100]	5	100	20	0.4	##	0.01
PSO	[0, 100]	[0, 100]	5	100	20	##	##	##
CS	[0, 100]	[0, 100]	5	100	20	##	0.25	##

Note: C is the penalty coefficient, which is the tolerance for errors. g is the kernel function of the model, which determines the distribution of the data after mapping to the new feature space.

**Table 2 sensors-23-01345-t002:** Extracted data parameters.

Indicators	Definition
Steering wheel angle (ratio)	The proportion of steering wheel angle that exceeds the set angle within a certain time window, where the set angle here is in the 75th percentile of steering wheel angle within the time window.
Steering wheel speed (ratio)	The percentage of steering wheel speed that exceeds the set speed within a certain time window, where the set speed here is in the 75th percentile of steering wheel speed within the time window.
Lateral offset distance (m)	The displacement of the main vehicle from the lane centerline within a time window.
Lateral speed (m/s)	The speed of the main vehicle traveling in a time window is decomposed into the magnitude of the speed in the normal direction of the road.
Lateral acceleration (m/s^2^)	The acceleration of the main vehicle traveling within a time window is decomposed into the magnitude of the acceleration in the normal direction of the road.
Longitudinal speed (m/s)	The speed of the main vehicle traveling in a time window is decomposed into the magnitude of the speed in the tangential direction of the road.
Headway distance (m)	The length of the distance between main vehicle and lead vehicle within a time window.
Throttle angle (ratio)	Percentage of the main vehicle’s throttle angle during a time window. When the brakes are not applied, the ratio is 0, when the brakes are applied, the ratio is 100%.
Longitudinal acceleration (m/s^2^)	The acceleration of the main vehicle traveling within a time window is decomposed into the magnitude of the acceleration in the tangential direction of the road.

**Table 3 sensors-23-01345-t003:** Results of ANOVA.

Indicator Variable	State of Driving	Mean	S.D.	F-Value	*p*-Value
Steering wheel angle	Baseline	0.186	0.334	73.486	0.000
	1-back	0.345	0.294		
	2-back	0.387	0.265		
Steering wheel speed	Baseline	0.118	0.153	16.816	0.000
	1-back	0.258	0.172		
	2-back	0.276	0.157		
Mean longitudinal speed	Baseline	20.04	1.26	5.457	0.004
	1-back	19.79	1.54		
	2-back	19.54	1.34		
Mean headway distance	Baseline	32.51	10.69	39.618	0.000
	1-back	37.49	13.41		
	2-back	38.09	13.01		
Standard deviation of headway distance	Baseline	0.94	0.98	10.592	0.000
	1-back	1.07	0.88		
	2-back	1.17	1.11		
Standard deviation of throttle angle	Baseline	0.084	0.096	26.232	0.000
	1-back	0.108	0.109		
	2-back	0.129	0.105		

One-way ANOVA significance level: 0.05.

**Table 4 sensors-23-01345-t004:** Classification accuracy of different optimization models.

Optimization Model	Parameter c	Parameter g	Accuracy of Cross-Validation	Accuracy of Classification
GA-SVM	32.19	4.38	93.96	94.01
PSO-SVM	2.11	3.74	93.16	92.28
CS-SVM	6.21	2.44	92.45	92.08

**Table 5 sensors-23-01345-t005:** Discriminant results of each model of cognitive distraction.

Indicators	Model	Accuracy (%)	Precision	Recall	F1 Score
Longitudinal and Lateral	GA-SVM	94.01	0.974	0.852	0.909
	PSO-SVM	92.28	0.937	0.841	0.886
	CS-SVM	92.08	0.958	0.804	0.894
	RF	91.19	0.926	0.801	0.858
Longitudinal	GA-SVM	76.62	0.767	0.698	0.731
	PSO-SVM	75.9	0.755	0.698	0.725
	CS-SVM	75.53	0.758	0.671	0.712
	RF	73.41	0.785	0.647	0.709
Lateral	GA-SVM	81.14	0.875	0.786	0.828
	PSO-SVM	80.78	0.821	0.792	0.806
	CS-SVM	80.26	0.804	0.766	0.785
	RF	78.32	0.756	0.751	0.753

## Data Availability

Data, models, or codes that support the findings of this study are available from the corresponding author upon reasonable request.

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
