# Peer review of "Young Novice Drivers’ Cognitive Distraction Detection: Comparing Support Vector Machines and Random Forest Model of Vehicle Control Behavior"

_sensors, 2023, doi:10.3390/s23031345_

Round 1

Reviewer 1 Report

The article uses the support vector machine and the random forest algorithm to compare the cognitive detection of the novice drivers, which is very meaningful, but there are some questions for the author to explain and reply:

 1. What is the difference between a young novice and a skilled driver?

2. Are the 36 experimental participants sufficient? Do male and female drivers need to be separated separately?

3. In this paper, it is not clear why only lateral lane keeping performance and longitude acceleration performance are extracted. These two indicators are added as cognitive distraction indicators. Are there any other representative indicators?

Reviewer 2 Report

The authors study on the young novice drivers’ cognitive distraction detection, which lay emphasis on comparing support vector machines and random forest model of vehicle control behavior. The research topic is very interesting and the articles are generally well written. Although I think the article has publication potential in SENSORS, there are some concerns.

1. Combined with the results of the literature review, the authors should further explain the unique value and significance of the targeted research on cognitive distraction recognition of young novice drivers for practical engineering applications.

2. As mentioned in Section 2.2, the simulated driving environment is four lanes in both directions with a speed limit of 80km/h. Please explain the reason for setting the virtual driving environment parameters in this way. Considering the current division method of highway and urban road and the practical application value of the research results, the basis of setting parameters of virtual environment should be explained in the research, and whether the virtual environment is close to a certain road environment in the real world.

3. In Section 2.4, the study participants were 36 drivers (licensed drivers) aged 22 to 26. Since the authors focus on cognitive distraction in young drivers, it is particularly important that the participants are representative of young people (especially novice drivers). The author should analyze the representative problem of the research object (or research data).

4. In Section 5, the authors should further discuss the research results, and should further reveal the deep conclusions (from the perspective of traffic safety or driving behavior) and the beneficial contribution of the research results corresponding to the research topic.

Reviewer 3 Report

This study developed driver’s cognitive distraction models using support vector machines and random forests based on vehicle control behaviors. The paper is well-written and their findings would be worth for the development of driver’s distraction detection systems in future research. However, there are some comments regarding the article.

The following comments are provided for the authors’ reference.

1.      [Abstract] Please clearly describe the objective of the study, add the number of participants, and state the implication of the findings obtained in this study.  

2.      [Introduction, page 2 line 97] Authors need to explain why this study used SVM and RF for the detection model of driver distraction. It is better to highlight the performance of those methods in existing studies so that readers could understand the state-the-art of the research.

3.      How to measure the vehicle control behavior? What kind of devices is used in this study?

4.      What device is used to perform the N-back task?

5.      [Methods] Please state the design of experiment (e.g., repeated within-subjects design, counterbalance scenario, etc.).

6.      [Methods] Please describe the driving performance measures (e.g., longitudinal velocity, steering wheel angle, etc.) in this section.  It is better to move section 2.6 in the method section to maintain the flow of the paper.

7.      [Participants] Please state the gender and standard deviation of the ages of participants. In addition, please add the approval number from Institutional Review Board (IRB).

8.      [Tables] Please check the font size for the text in the all tables to match with the journal requirements.

9.      [Page 9, line 241; Page 9 line, 251, etc.] Please rewrite the ANOVA results following the APA format. There are something missing in the current reporting style.

10.   [Figure 7, 8, etc.] Please highlight the significance (e.g., using Tukey test) among those levels (baseline, 1-back, and 2-back) in all graphs.

11.   [Figure 7-11] Please match the Y axis unit with the table 3.

12.   [Section 2.6] What is the significant levels of ANOVA? Is it a within-subject design?

13.   [Table 5] Please highlight in bold the results with high performances.

14.   [Discussion] The discussion section is too short. Please separate the Discussion and Conclusion section. Please more elaborate on the findings of this study compared with existing studies in this section. In addition, it is better to explain some trends obtained in this study in the Discussion section. Limitation and future work is not mentioned clearly.

15.   Please check the mismatch use of capital letter.

Round 2

Reviewer 1 Report

The authors have made considerable effort and addressed my concerns.

Reviewer 3 Report

The authors have revised the manuscript based on the reviewer suggestions. I have no further comments.